# “We Just Get Whispers Back”: Perspectives of Primary and Hospital Health Care Providers on Between-Service Communication for Aboriginal People with Cancer in the Northern Territory

**DOI:** 10.3390/cancers17193155

**Published:** 2025-09-28

**Authors:** Emma V. Taylor, Amy Elson, Bronte Avishai, Philip Mayo, Christine Sanderson, Sandra C. Thompson

**Affiliations:** 1Western Australian Centre for Rural Health (WACRH), University of Western Australia, P.O. Box 109, Geraldton, WA 6531, Australia; sandra.thompson@uwa.edu.au; 2Community Allied Health & Aged Care Services, NT Health, 258 Trower Road, Casuarina, NT 0811, Australia; 3Alan Walker Cancer Care Centre, Royal Darwin Hospital, 105 Rocklands Drive, Tiwi, NT 0810, Australia; 4Menzies School of Health Research, Charles Darwin University, P.O. Box 41096, Casuarina, NT 0811, Australia; 5Territory Palliative Care—Central Australia, Alice Springs Hospital, Alice Springs, P.O. Box 2234, Alice Springs, NT 0871, Australia

**Keywords:** Indigenous, Aboriginal and Torres Strait Islander, First Nations, cancer care coordination, communication, cancer services, primary health care, health professionals

## Abstract

Aboriginal and Torres Strait Islander people have worse outcomes from cancer compared to non-Aboriginal people, particularly in remote areas. In the Northern Territory (NT), Australia’s third largest state/territory, cancer is one of the main causes of death for Aboriginal people. Primary health clinics play an important role in cancer screening, diagnosis and cancer care, particularly in remote communities, so accurate, timely communication between clinics, hospitals and specialists is essential. This qualitative study interviewed 50 staff from 15 health services across the NT and found that poor or delayed communication between health services negatively affected patient care and support. This research highlights the need for more timely communication and improved health care IT systems to enable better information sharing between services. It also identifies the need for designated clinic staff to support patients with cancer and dedicated Aboriginal cancer roles to improve the coordination and delivery of care for Aboriginal cancer patients.

## 1. Introduction

Aboriginal and Torres Strait Islander peoples in Australia experience poorer cancer outcomes compared to other Australians; a disparity that has been observed in countries with Indigenous populations worldwide [1,2]. ‘Aboriginal’ is the term preferred in the Northern Territory and therefore is used in this paper to refer to the Aboriginal and Torres Strait Islander people who are traditional inhabitants and Indigenous people of Australia. ‘Indigenous’ has been used when we refer to the First Nations inhabitants of other countries. In the Northern Territory (NT), Australia’s third largest state/territory and the least populous, cancer is a leading cause of death for Aboriginal peoples with a persisting gap in cancer mortality rates between Aboriginal and non-Aboriginal populations [3,4,5]. Many factors contribute to these poorer cancer outcomes, including reduced access to health services, lower participation rates in national cancer screening programs, later stage at diagnosis and lower uptake and completion of cancer treatment [6,7,8]. With 74% of Aboriginal people in the NT living in remote or very remote areas, the need to leave Country and travel long distances to access care in a health system which is not always seen as culturally secure presents well-recognised challenges [9,10].

Cancer care is complex and often involves multiple different health care services, a complexity that increases with remoteness [11,12,13]. Delayed, absent or poor communication between health care professionals is a global challenge that may negatively impact continuity of care and patient outcomes [14,15]. Primary health care (PHC) clinics often act as health hubs in remote communities around the world and have an important role in cancer screening, diagnosis, and post-discharge cancer care, where patients are further from specialist cancer services, so accurate communication between hospitals, specialists and PHC clinics is vital [16,17,18]. Improving the coordination of cancer care is critical for addressing rural and remote patients’ needs and improving their cancer outcomes [19,20], and particularly salient for rural and remote Indigenous patients [21,22,23].

Vast distances, low population density, and a more transient health workforce in the NT present major challenges for health service and specifically cancer treatment delivery, amplifying the importance and challenges of coordinating cancer care [24]. The difficulties of geographical isolation are exacerbated in some remote Aboriginal communities, where lack of access to transport and poor internet or telephone reception may require clinics to operate as health communications hubs, contacting patients for specialists, organising specialist appointments and telehealth consults. Challenges in coordinating care to meet Aboriginal cancer patients’ needs include lack of communication between services, delays in receiving timely hospital information, distance to treatment and provision of cultural support [10]. These challenges have also been reported by health service providers servicing rural communities in other parts of the world including in the United States of America (USA) [25,26,27,28], Ethiopia [29] and Scotland [30] and in rural and remote Indigenous communities in New Zealand [31] and Canada [13].

Two studies from Queensland have examined communication between PHC services, hospitals and specialists in relation to providing cancer care for Aboriginal people [16,32]. A retrospective audit of clinical records by Valery and colleagues found that over two-thirds of patients who visited a PHC service for cancer-related care did so before a hospital discharge summary was available, and that over 40% of discharge summaries lacked details regarding discharge medications [16]. De Witt and colleagues interviewed 17 staff from Aboriginal Community Controlled Health Services (ACCHSs) and 9 staff from a large urban cancer treating hospital and found that lack of communication and timely information being received by the PHC service from the hospital were core challenges providing quality cancer care [32].

Minimal research has explored the perspectives of both primary health care providers based in remote and very remote locations and hospital health care providers on between-service communication, particularly for Aboriginal cancer patients. Primary care and hospital-based health professionals, with their knowledge of the health system’s inner workings and complexities, are well placed to identify gaps in communication and cancer care coordination, as well as opportunities for improvement. In addition, health professionals working in remote areas have a unique understanding of the complexities of providing health care in a remote community, and the additional challenges of communicating with their hospital-based colleagues. To our knowledge, no such study has been conducted in the NT.

This qualitative study was undertaken to investigate the perspectives of primary care providers based in regional and remote communities of the Northern Territory as well as the perspectives of hospital-based health care providers on communication issues between these services. It also reports recommended strategies to improve communication and enhance the delivery of cancer care for Aboriginal people.

## 2. Methods

This study employed a qualitative design, drawing on semi-structured interviews with health professionals. Participants were recruited from PHC services in regional, remote and very remote geographical locations, as well as from hospital and cancer services in regional and remote locations. This qualitative methodology facilitated an in-depth exploration of health professionals’ experiences and perspectives on the provision of cancer care to Aboriginal patients. An article reporting on health care providers’ perspectives on provision of cancer screening to Aboriginal people in the NT has previously been published from this research study [33].

### 2.1. Ethics Approval and Project Background

Ethics approval to conduct this study was granted by the Human Research Ethics Committee at The University of Western Australia (RA/4/1/6286) and the Human Research Ethics Committee of the Northern Territory Department of Health and Menzies School of Health Research (2016-2576).

This study was started as part of a national initiative aimed at identifying and describing cancer services providing treatment to Aboriginal cancer patients across Australia. It was a component of research conducted under the auspices of the Centre for Research Excellence (CRE), Discovering Indigenous Strategies to improve Cancer Outcomes Via Engagement, Research Translation and Training (DIS-COVER-TT) funded by the National Health and Medical Research Council (NHMRC). The CRE was led by an Aboriginal researcher and brought together Aboriginal and non-Aboriginal researchers from around Australia, with the aim of improving outcomes and services for Aboriginal people with cancer. The study was carried out in accordance with the NHMRC Guidelines for Ethical Conduct in Aboriginal and Torres Strait Islander Health Research [34].

### 2.2. Research Team

The research team involved in the analysis consisted of six researchers. Three of the non-Indigenous team members have clinical backgrounds and extensive experience in cancer care in the Northern Territory (AE, BA and CS). PM is of English, Welsh, Aboriginal and Torres Strait Islander heritage, with ancestral ties to the Gurindji and Mudburra peoples of the NT, as well as to the Torres Strait Islander peoples of the Badu and Mabuiag Islands in Queensland. PM is based in the NT with extensive experience providing health care for Aboriginal peoples. SCT and EVT have over 30 years combined experience with collaborative research into improving Indigenous health outcomes.

### 2.3. Participants and Recruitment

Participants were purposively recruited from PHC clinics in regional, remote and very remote NT locations, from the regional cancer centre, and from regional and remote cancer-treating hospitals in the NT. Aboriginal and non-Aboriginal health professionals and support staff were eligible if they were involved with the treatment, care or support of Aboriginal cancer patients, or if they filled a leadership role in the care of Aboriginal cancer patients.

Potential staff participants were invited to take part in the study through personal approaches in clinical service areas and at staff meetings. Study information was provided either in-person or via email. Some participants forwarded the study information to colleagues or recommended additional participants to the research team. As data collection progressed and gaps in representation were identified, recruitment efforts shifted to focus on including participants whose perspectives were underrepresented. Participation was voluntary and all participants provided written or oral consent before being interviewed.

### 2.4. Data Collection

Semi-structured interviews took place between September 2016 and April 2019 and were conducted by a highly experienced remote area nurse. A semi-structured interview guide was developed by the investigators, reviewed by the Aboriginal Advisory Group and refined. Most interviews were conducted in-person at the participant’s workplace; however, a small number were conducted via telephone due to issues of availability and geographic distance. The interviews ranged in duration from 20 min to 2 h and 45 min, with most taking slightly over one hour. All interviews were audio-recorded with the participants’ consent, except for one interview where the recording equipment failed and the extensive notes taken during the interview were used. Interview recordings were transcribed verbatim by two team members and checked against the recordings for accuracy.

The interview guide (Appendix A) included open-ended questions on the participant’s role in delivering cancer treatment providing care to Aboriginal people, the typical treatment pathway for Aboriginal cancer patients, communication between primary and tertiary health services, as well as questions around referral, diagnosis, treatment, discharge and post-treatment care, end-of-life-care. Participants were also asked about ways in which their health service and cancer services generally could improve to better meet the needs of Aboriginal patients. The Australian Statistical Geography Standard (ASGS) Edition 03 [35] was used to categorise the geographic remoteness of the NT health services and participants as outer regional (Darwin and surrounds), remote or very remote (Figure 1).

### 2.5. Data Analysis

We followed the qualitative data analysis process described by Green et al. [37] of immersion in the data with rereading, coding, categorisation, and aggregation of identified themes. Data analysis was delayed due to the outbreak of COVID-19 and changes in the project team. In 2023, transcripts were de-identified and imported into NVivo 13 for initial data organisation. Transcripts were reviewed by three team members. Analysis was iterative, with coding undertaken through line-by-line review of the transcripts by one researcher (EVT) to identify emerging themes.

During this analysis, between-service communication was identified as an area of interest due to the rich data collected. Two researchers (EVT and AE) re-examined the existing themes to identify and refine themes specifically focused on between-service communication. AE analysed the data to identify barriers and enablers to communication as well as recommendations to improve communication. Coding discrepancies were discussed between AE and EVT who refined themes and triangulated staff interviews. BA and PM then reviewed a sample of the coded transcripts and the emerging themes related to between-service communication to ensure that themes were consistently applied and meaningful in the current context. Analysis was documented to ensure that each theme could be traced back to the original data, using direct references from the interviews to provide evidence and maintain the voice of the participants. Recommended strategies were then grouped according to which barrier or enabler they primarily addressed, even though some strategies are relevant in addressing more than one barrier. Reporting of the findings was guided by the consolidated criteria for reporting qualitative research (COREQ) checklist [38]; checklist available on request.

Between May and June 2025, the findings and recommendations were presented to 22 key stakeholders to confirm their contemporary relevance. Regional, remote and very remote clinicians (*n* = 11) involved with cancer care for Aboriginal people in the NT were approached and invited to provide feedback on our findings. In addition, our findings were presented at a team meeting of health service managers (*n* = 11). After presenting our findings to the stakeholders, their feedback and reactions to the findings were captured through extensive note taking. The stakeholders agreed with the identified barriers, enablers and recommendations and confirmed that the findings were still highly relevant. The views of the stakeholders are not reported in the Section 3.

In addition, the stakeholders were asked whether they had any additional recommendations to improve between-service communication, not already identified in our findings. The stakeholder recommendations were mapped to the existing recommendations, and new strategies were grouped according to which barrier or enabler they addressed. Recommendations based on strategies suggested by interview participants and key stakeholders are reported in the Discussion Section 4.2 of this paper and any recommendations solely provided by the stakeholder group are identified.

## 3. Results

Fifty participants were interviewed from 15 health services. Health services consisted of primary health care (PHC) clinics (*n* = 8), hospitals (*n* = 3), a cancer centre (*n* = 1), and support services (*n* = 3). Most of the services (*n* = 10, 66%) were located in remote or very remote areas, with the remaining third (*n* = 5) located in the outer regional area of Darwin and surrounds. All PHC clinics were operated by NT Health.

Almost half the participants (*n* = 23, 46%) lived and worked in a very remote location, with 13 (26%) living in a remote area and 14 (28%) living in the outer regional area of Darwin and surrounds (Table 1). Participants included more women (*n* = 38, 76%) than men, 33 (66%) were aged between 40 and 59 years of age, and 14 (28%) identified as Aboriginal or Torres Strait Islander. Over half (*n* = 27, 54%) had been in their position less than 5 years (range 3 months to 33 years). Participants had on average 10 years of experience providing care to Aboriginal patients (range 6 months to 33 years) and two-thirds (*n* = 33, 66%) worked in primary health care. A diverse set of professions participated, including Aboriginal Health Practitioners (AHPs), Aboriginal Liaison Officers (ALOs), registered nurses (RNs), general practitioners (GPs), physicians, social workers, dietitians, managers, and administrative staff. Some participants held dual roles (e.g., cancer care coordinator and palliative care nurse).

Analysis of the interviews revealed several enablers and barriers to communication between rural and remote PHC clinics and regional and remote hospital services about the Aboriginal cancer patients they cared for (Figure 2).

### 3.1. Communication Barriers

#### 3.1.1. Fragmented and Inefficient Information Systems

Poor flow of information was perceived as the biggest barrier to communication between health services. As summarised by an RN in a very remote community, *“It is about communication. Our systems don’t talk to each other, and our people don’t talk to each other”* (Very Remote PHC RN HP029).

Siloed Health Care IT Systems

A major barrier to communication, perceived by hospital and PHC staff (*n* = 11), was that hospitals and PHC clinics in the NT use different IT systems to store and transmit patient records. These systems don’t easily interface, making it difficult to share records between services or for staff in one service to access patient records in a different service. This resulted in delays in accessing important medical information such as patient notes, test results and prescriptions. Staff reported patients being lost to follow up care because they couldn’t track if or when patients presented at other health care services. As the Cancer Centre Manager explained, *“The patient gets lost in the system… because we don’t have electronic record system where we can track patients from the community to the acute setting”* (Regional Cancer Centre Manger HP001).

These IT system communication issues were exacerbated for remote hospitals and PHC clinics located near a state border who often saw patients from different states, or who had patients travel to South Australia for treatment. As one RN in a very remote community explained:


*“It’s very difficult because we use PCIS [Primary Care Information System] and the hospital doesn’t, we can’t see what’s in the hospital records and they can’t see what’s in ours… everyone uses different systems… we are so close to the border a lot of people here come over from [remote communities] in Queensland, and so it’s all that stuff to deal with too and that’s a whole different system as well”*
(Very Remote PHC RN HP017).

Disjointed, Delayed and Missing Information

Another major barrier raised by hospital and PHC staff (*n* = 11) was delayed or absent communication between health services. Staff at six remote or very remote PHC clinics reported that they didn’t get information back about cancer patients, giving examples of being unaware of upcoming patient appointments. This was particularly problematic when the clinic needed to notify the patient or make travel arrangements for the patient to attend, support upcoming telehealth appointments (that the clinic needed to notify the patient about or organise), patient test results or diagnosis, and not receiving discharge summaries in a timely manner. Delays and absence of information flowing from hospitals to PHC clinics were also observed by two remote hospital staff, although this was not identified by regional hospital staff.

The delays and lack of communication occurred across multiple points on the cancer patient journey, with PHC staff providing multiple examples of information not being received or acted upon in a timely manner. Negative outcomes of this related to patient travel arrangements not being made, patients missing appointments (both in person and telehealth) and patients disengaging with follow-up treatment. PHC staff often lacked information needed to explain treatment and why it was important. Communication issues and delays were exacerbated when the patient had travelled interstate (usually from Central Australia to Adelaide) for treatment.


*“We fax the referral off, and we will let [the patient] know when an appointment comes. And then, so the whole system can fall down because did we fax it off? Did it get received? Often not. Doesn’t seem to get received. If I look up on the computer, I can see if someone’s got an appointment for 10 months in the future but they don’t tell our practice manager until about 3 weeks out. The patient still doesn’t know. And so if we don’t get that notification, or it gets lost in the email trail, or we are busy, then the travel arrangements can’t be made in time”*
(Very Remote PHC GP HP043).

A very remote community RN described patients disengaging with care when PCH staff weren’t in the loop about appointments and the reason for recalls:


*“They [the patient] go to their outpatient appointment. Do you get something back from the outpatient appointment? Quite often, particularly if it’s a routine following on thing… you don’t get anything back. I’ve rung up and outpatient people have been really helpful and they’ve looked in notes and said ‘Oh doctor has written see in 2 months’ or something like that. You don’t necessarily get a letter back from the outpatient appointment… So another letter comes—so you say, ‘Will you go to that again?’ And they [the patient] say ‘What for? I’ve been.’ And if you don’t really know why they’ve been, then you’ve got to start the whole process again”*
(Very Remote PHC RN HP032).

Delayed or missing discharge summaries were described as “ridiculous” barriers by staff at PHC clinics. This resulted in patients not being followed up by PHC staff who were unaware that patients had completed their treatment and returned to community and PHC staff being unable to provide appropriate medical care because they were missing critical clinical information. An RN at a remote community describes the problem:


*“A load of stuff happens [when the patient is in Darwin for diagnosis], and we never ever hear back what happens or and then they get sent back to community and then we go ‘oh crap what are we meant to be doing?’ Our patients… come back and there’s no plan, what’s happening? We don’t get a lot of feedback from town… Even when they get off the plane, when they are in community and we are like “wow are they back?” They were probably back a week ago and we haven’t sort of followed them up because we didn’t know they were back”*
(Remote PHC RN HP015).

PHC staff also commented on missing information in the referral or discharge summary, including lack of contact information. One RN at a regional clinic described how they had to become “an advocate” for their cancer patients, chasing up information such as post-treatment medication details, made more challenging because there was no contact information for the discharging service.

The Cancer Care Coordinator at one remote hospital observed that one cause of delays or absence of information was “a lot of patients” not having a GP which resulted in hospital staff not knowing who to send information to. Regional hospital staff commented that some patients don’t have a fixed address or move between multiple communities, however the system only allows for one patient address making it challenging to know which clinic to send information to.

One staff member at the regional cancer centre reported a lack of communication from the PHC clinics, expressing frustration at PHC staff not replying to e-mails or actioning e-mails in a timely manner which resulted in delays and patients missing appointments.

This lack of communication both from hospitals and PHC clinics, resulted in both hospital and PHC staff being uncertain about what happened to cancer patients after referral or discharge to another service. One very remote community RN described the disjointed flow of information, *“Once they go off to Darwin, half the time we don’t know, we just get whispers back”* (Very Remote PHC RN HP017). A regional Cancer Care Coordinator echoed this sentiment:


*“Do we ever see [the patient] again?… I don’t know, do we keep records of that? How do they go with getting some palliative care? If they need it in the community once they’ve gone back, is their pain well controlled? We let them go and then that’s it and I don’t know”*
(Regional Cancer Care Coordinator HP006).

Information flow and communication were not reported as a barrier by all remote PHC staff. There was diversity of opinion about the efficacy of communication between services, even within the same PHC clinic. Three PHC staff felt they received sufficient and timely communication from cancer services, particularly Alan Walker Cancer Centre and palliative care services, and several regional hospital staff commented on how well they communicated with PHC clinics.

#### 3.1.2. PHC Staff with Limited Knowledge of Cancer

Multiple hospital and PHC staff believed that PHC staff didn’t have a good understanding of cancer treatment or cancer pathways, which PHC staff stated was exacerbated by a lack of information from the hospital. PHC staff didn’t know who to contact at the regional hospital to get information about their patient or for their patient. This was attributed to cancer being complex, with multiple and changing treatment pathways. As one RN in a very remote community described, *“What I find really difficult out here is finding the services I need to access and who you need to talk to. It’s not easy. I’ve basically stumbled on it. Or talking to people. ‘How do I do?’ ‘Where do I get this?’ Because there are no clear pathways.”* (Very Remote PHC RN HP017) Referral and treatment pathways were particularly complex for less common cancers with which PHC staff had less familiarity. This resulted in PHC staff being less confident in communicating with patients about what to expect because they didn’t have enough information. In the words of the Cancer Centre Manager, *“they are not well informed to inform the patient”* (Regional Cancer Centre Manger HP001). A GP in a very remote community agreed, and felt communication could be improved by more information from the regional hospital.


*“I think probably as GPs [we] don’t understand the process because… like we get a letter. We understand it’s cancer. We understand that they are going to need A, B and C. But how that actually works… that’s not clear… The pathway isn’t clear, even to the doctors… It would be great to have two letters. The letter that says this is the diagnosis and please talk to your patient about this, and then the letter that says here’s the next steps.”*
(Very Remote PHC GP HP027).

More information from the regional hospital was required during treatment and at discharge to help PHC staff provide care for chemotherapy and radiation therapy patients. PHC staff themselves noted that being poorly informed and not educated on how to care for patients during and post- cancer treatment could lead to worse outcomes for the patient if staff were unaware of the clinical significance of certain symptoms or medication interactions. *“The worst part is no discharge summary…. Like a lot of rural nurses, I’ll be honest with you… they are brilliant at what they do. But they have no idea about after care with chemo or radiation. They have no idea what to look out for”* (Very Remote PHC RN HP017).

#### 3.1.3. High Turnover of PHC Staff

High PHC staff turnover was seen as a substantial communication barrier by hospital and PHC staff. Hospital staff reported that it was challenging to know who to contact about patients. Both hospital and PHC staff perceived that new staff might not know which patients required following up, resulting in “a lack of continuity of care”.

“*Every time I phone a clinic, I’m talking to a different nurse, or a different GP who’s covering and I’ve got to go through the whole history of the patient because they don’t know who they are. Their time is limited. They’re only there for two weeks”*(Remote Hospital Cancer Care Coordinator HP049).

High turnover resulted in delays in patients being followed up with results and planned cycles of review post-treatment getting “lost in the system”. Locum and agency staff (common in remote clinics with high turnover) were more focused on acute care rather than going through the system and actively seeing which patients needed following up. A GP described the situation at their very remote community:


*“Locums, they are here for a short period of time and they kind of manage what walks through the door and what people bring to them you know. They are probably not in a position to be kind of searching through and actively seeking things…. When I started here there was stuff in the inbox from [2 years ago] and I think what happens, you just don’t know what to do with that. So I’ll leave it there so it doesn’t get lost but I don’t know what to do with it. And the next locum comes and goes, that’s from [2 years ago] that can’t be relevant…. Or you have an agency nurse that takes the recall off [for a routine mammogram or CT scan to monitor a lung lesion] because they didn’t think that the patient attended so the recall gets disappeared. I can’t remember a thousand people’s needs. You know there’s lots of steps along the way where it’s really easy for the non-urgent stuff to disappear.”*
(Very Remote PHC GP HP027).

#### 3.1.4. Regional Staff with Limited Understanding of Remote Health Care Challenges

RNs and GPs at four remote and very remote clinics believed that regional hospital staff had limited understanding of the limitations and challenges faced by remote PHC clinics, particularly lack of understanding regarding the availability of tests, medicine, equipment, and ability to contact patients. Regional staff giving insufficient advance notification to PHC staff about requirements negatively affected patients, with PHC staff giving examples of delays in acquiring prescribed medications that weren’t readily available in remote communities and delays with patient diagnosis and treatment caused by difficulties in organising or sending tests. An RN in a very remote community gave multiple examples:


*“I need advanced warning if you want a specialised test. I really do, ‘cause I only send bloods out on a Tuesday and a Thursday. And if it has to be frozen or something weird you need time to sort it out. Because I might not even have the tubes to be able to run that test…*



*We had a gentleman and… he got sent back with a pigtail drain in, with bags attached and having to change them…. we don’t have access to that stuff. You need to send it with him, enough to get him through, because we don’t have it. Or tell us what he needs before you send him back.”*
(Very Remote PHC RN HP017).

PHC staff felt regional staff didn’t understand the complexities of contacting very remote patients, who may not have a phone or a fixed address; this may require a PHC staff member to drive a long distance to contact them or find them. An RN from a very remote clinic highlighted this when describing an interaction with a nurse from the regional hospital:


*“She [the breast care nurse] thought just ringing someone up and telling someone to do this, it would magically happen. Whereas it actually meant somebody had to actually go out to an outstation and find this women and tell her, she had to come in and have her bloods done in time to catch the airplane. The bloods go on a plane and get results and [patient] has her appointment after that. And [the breast care nurse] didn’t understand”*
(Very Remote PHC RN HP032).

### 3.2. Enablers to Communication

#### 3.2.1. Persistence and Personal Connections

Individuals who used system workarounds or put in considerable additional effort to improve communication with health services or obtain important information about patients saw themselves as enabling communication between services. Multiple hospital and PHC staff described their persistence with contacting multiple staff and health services, noting that if they didn’t go to those lengths the information would not get through (hospital staff) or they would lack information they needed to care for their patients (PHC staff). *“The [cancer] pathway is person dependant… you can have great systems set up, but you need the person, they are like the oil in the systems, so if you don’t have that person that maintains the systems, a system will fall over”* (Very Remote PHC RN HP032).

Two hospital staff described their efforts to share information with PHC clinics. The Cancer Care Coordinator at a remote hospital described how she and the Resident Medical Officer (RMO) had taken on relaying timely information to PHC clinics due to the “tardy” timeframe for letters to be dispatched. The ALO at the regional cancer centre described the multi-step process they followed to send any available information to the relevant PHC clinic when an Aboriginal cancer patient was discharged to a remote community:


*“If there’s any paperwork I can forward to the clinic, I will print them off, scan them and email them to the Community Health Clinic… If there’s a discharge summary, or some sort of follow-up letters, or anything I can find that I can have access to, I’ll send it off to them. It’s not my role to do that, but I do it… I feel there’s a need because I think communication between us and the community… I understand the isolation”*
(Regional Cancer Centre ALO HP007).

The following vignette (Box 1) describes the difficulties an RN in a very remote community encountered finding out what happened to a patient she had organised medical evacuation for thinking the patient had a terminal abdominal tumour. It illustrates the way individuals go beyond their role to support their patients and facilitate communication despite the barriers.

Box 1Vignette[The patient] turned up again, about 8 weeks
later. She came in to see me because she wanted a new dressing because she
had a cut right up and down her tummy. I was saying, “It’s lovely to see you. You can
have a new dressing. What happened?” She said “They fixed it, it’s all good. I’m
good Sister”. I’m thinking that she had been sent back
palliative. So I’m trying to find out what is going on. So that was a
torturous journey in itself. I rang Alice Springs. She had been sent to
Adelaide to one hospital, but that hospital had sent her to another hospital,
but the doctor that had cared for her was now working at a 3rd hospital.
Nobody could actually find any notes. I tracked down the doctor who remembered her
very well. She did have a tumour. I don’t remember its name, but it’s
relatively rare. It’s one that grows from the kidney, if you take the kidney
away, you have a really good prognosis. And it was as big as a football
apparently. She needed units and units of blood. It was only that I was persistent and I rang
these 3 hospitals and actually tracked down the doctor that looked after her.
The doctor said, “Well we sent her notes to
oncology in Alice Springs”. But that’s sort of a visiting service. I don’t
know how well they communicate. But nobody there seemed to think it was
necessary for the clinic that I was working in to know anything about what
happened to this lady. They couldn’t understand why I would like her notes or
a letter or whatever…They are saying “She’s fine, she’s well. We’ll
send an outpatient appointment in 6 months or so”. I had put her on a plane with her family and I
was suspecting to myself ‘I’ll never see you again’. And they couldn’t
understand why the clinic would need any notes. And what about the significance of having only
one kidney now? You’d be very careful about Methotrexate drugs that you give
knowing she’s only got one. You know I’d like to put an alert up that she
only has one kidney. I found that really quite remarkable and
bizarre… I’m presuming that Adelaide sent some notes back to Alice
Springs Hospital but it was practically impossible to find this out. It was
just that I was persistent and determined.(Very Remote PHC RN PH032)

#### 3.2.2. Specific Roles Facilitating Communication

Several roles were identified as facilitating communication between health services; PHC clinic managers, Breast Cancer Nurses (BCNs) and Aboriginal Liaison Officers (ALOs) were all specifically mentioned.

PHC clinic managers were regarded by both hospital and PHC staff as key in coordinating and facilitating communication between the hospital and patient. Hospital staff relied on clinic managers to pass on information to patients—particularly those without a mobile phone, who had poor phone reception or didn’t have a fixed address. Clinic managers were described as “paramount” to the successful coordination of telehealth appointments between hospital staff and patients and their families.

The Breast Cancer Nurses (BCNs) were reported to improve communication between the regional health service and GPs based in remote PHC clinics, because GPs were aware of the BCNs and felt comfortable contacting them directly. GPs stated that as breast cancer was a more commonly occurring cancer in their patients, they had greater familiarity with the patient pathway and who to contact. GPs were comfortable contacting the BCNs directly to discuss which diagnostic tests to organise and help plan for the patient. This streamlined care and helped emphasise that *“[a] person has been booked and needs to be followed up so that they don’t disappear”* (Remote PHC GP HP013).

ALOs were a central, visible point of contact between the hospital and PHC clinics. One very remote RN described how the ALO would get in contact via e-mail to send discharge information, or information about a medication that the patient required or diagnostic tests that were required prior to the patient being seen.

#### 3.2.3. Telehealth Between Services

Regional staff reported that using telehealth improved communication with remote PHC clinics as well as with patients. Telehealth was described as building “stronger linkages” between the regional hospital and PHC staff and resulted in regional and remote staff talking to each other more and gaining familiarity by seeing each other on screen. Telehealth consultations with PHC clinicians sitting in on the appointment resulted in everyone being on the same page and allowed for a plan to be developed with the involvement of the PHC clinician, the patient and their family. A Cancer Centre Manager also noted that telehealth can be used to educate remote clinicians *“on what to do next”* (Regional Cancer Centre Manger HP001).

#### 3.2.4. Centralised Cancer Care Service

The Alan Walker Cancer Care Centre (AWCCC) was seen to facilitate communication through its efforts to make the centre approachable by clinicians from remote communities and building a reputation as a *“one stop shop”* (Regional Cancer Centre Manger HP001). The AWCCC Manager described their aim as *“all practitioners, GPs, nurses, anywhere, anybody can actually call us and talk to a specialist straight away”* (Regional Cancer Centre Manger HP001). This perception seemed to be flowing out to some remote PHC clinics, with nurses at two remote Top End clinics (but none in very remote clinics or Central Australia) reporting good communication and accessible staff at the cancer centre. As a Nurse Practitioner at one remote PHC clinic described, *“There seems to be fairly good communication from the Alan Walker Centre about where people are up to. They seem to know their client load really well… you put the referrals in and it gets followed up”* (Remote PHC Nurse Practitioner HP014).

## 4. Discussion

This paper examines health provider perspectives on between-service communication while caring for Aboriginal cancer patients in the NT. Between-service communication is not occurring consistently or in a timely manner, and while individuals are working to improve communication, multiple barriers exist. Fragmented and inefficient information systems with IT systems that don’t interface and delayed or missing patient information including discharge summaries present major barriers to care. PHC staff’s limited knowledge of cancer treatment pathways affected the care they could deliver. This was exacerbated by high turnover of remote PHC staff and regional staff with limited understanding of the challenges of delivering health care in a remote community. While some staff made big efforts to improve communication or track down patient information, a situation where communication efficacy depends on individuals (as opposed to systems or processes) results in care failure and inconsistencies. Needing to chase patient information places a substantial additional workload on stretched health care staff.

Poor communication adversely affects patient care for vulnerable people around the world [16,17,39]. Delayed or missing communication led to patients missing appointments, with services not informed of appointments or not informed in time to contact patients or organise travel. This is a particular issue for cancer patients, where even a four week delay in commencing treatment can lead to worse outcomes [40]. Staff gave numerous examples along the treatment pathway where communication breakdowns or delays led to patients falling through the gaps, being “lost” to follow up or disengaging from treatment, an issue that has long been reported both in Australia [41] and internationally [29,42]. One example raised by participants was patients not having a GP; however, it should be noted that some remote patients may not be familiar with the term “GP” due to cultural differences in health care terminology and remote clinics with high turnover of staff or predominantly staffed by nurses. Terminology such as “which clinic do you attend?” may be more readily understood and preferred by some patients.

Inadequate communication and lack of timely information presents a major challenge in delivering quality care to Aboriginal people with cancer, with the lack of information disproportionately affecting PHC staff. These findings are consistent with other studies, both in Australia and internationally, which found that limited or delayed communication between specialist and primary care negatively affected clinical decision making and advice provided by PHC clinicians, with rural and Indigenous patients particularly affected [16,29,39,41,43]. Improving timeliness of communication between hospitals/specialist and PHC clinics across the patient cancer pathway is vital, and could include updates to PHC during patient hospitalisation, timely information around hospital discharge for follow-up care, more efficient administrative processes and improved sharing of patient records and care plans [32]. An international narrative review found enablers for achieving continuity of care for First Nations people included information shared among providers and settings to ensure “collective memory”, links and referral strategies for care professionals and shared synchronized care records [39].

An integrated health care IT system that allows hospitals and PHC sites to share information and electronic patient records was seen as critical by participants and stakeholders. This need is consistently identified both within Australian and internationally, so that streamlined health IT systems allow for cancer patient’s records and care plans to be shared across services [32,44]. As a remote patient’s cancer journey involves accessing multiple health services across different locations, health professionals require access to patient records to provide continuous, coordinated care and increase the likelihood of improved patient outcomes [41]. However, a Canadian study cautions that implementing such a system can be challenging due to the technical challenges involved in developing and maintaining complex health information systems, especially when multiple digital interfaces must be coordinated to streamline workflows [44].

When the first interviews for this study were being conducted, the NT government announced the development of a digital health system called Acacia, touted as a replacement for NT Health’s “obsolete major systems” and integrating a number of clinical systems to “deliver a single, secure, electronic health record across hospitals, primary health care centres and community health services” [45]. In late 2020, the Acacia Read-only Electronic Patient Record was launched, and in 2022 installation began across hospitals in the NT. However, while stakeholders report that the system has made some improvements and “has potential”, the rollout has experienced numerous delays and setbacks [46]. At the time of writing, the system was still in the process of being implemented in hospitals across the NT, with two hospitals yet to receive the system; the timeline for rolling out the remaining stages including deployment to primary care clinics is unclear [46]. A system that connects all providers, including non-government-run clinics, and allows them to share patient information, is still urgently needed in the NT.

Around the world, the role of primary care in cancer diagnosis and management is growing [47,48]. Cancer is complex to manage, with numerous pathways depending on cancer type and stage [48] and these complexities are exacerbated for remote PHC staff. Criticisms that some PHC clinicians weren’t well informed of the appropriate cancer pathway or how to care for cancer patients, adversely impacting the care provided to patients may be justified but reflects a failure of communication and issues related to staffing, including training, retention and support. This mirrors findings from an Ethiopian study on the coordination of cervical cancer care, which reported that lack of feedback on referred patients hampered learning for PHC providers and secondary hospitals, as they were unsure if their cancer screening referrals were appropriate [29]. While the Optimal Care Pathways (OCPs) [49] aim to guide health professionals by providing guides for delivering best practice cancer care for more than 25 cancer types, additional education and guidance is needed for remote PHC staff [32]. A new OCP for people living in rural and remote areas is currently being developed [50], however patient-specific information and guidance from hospital staff and specialists is urgently needed by PHC staff at point-of-care. Given the high turnover of remote PHC staff in the NT [51], there is a need to rethink support systems to ensure staff know who to contact regarding patient care. Recommended strategies include having a dedicated contact person at each PHC and hospital site for health professionals to discuss cancer patient care, dedicated roles to support patients with cancer and training AHPs in cancer care [29,32]. Regional hospital staff need opportunities to experience the challenges of remote PHC staff and the realities of remote community life; cultural immersion experiences may help to improve the timeliness of communication to PHC services.

Several roles were highlighted as facilitating communication between health services, with PHC clinic managers, BCNs and ALOs all specifically mentioned. Since the interviews, the number of cancer care coordinators has expanded from two McGrath breast care nurses to multiple cancer care coordinators across specific types of cancer and multiple sites. This increase has almost certainly improved between-service communication because cancer care coordination has been found to improve organisation of patient care and increase effective and timely communication between health professionals [52]. The importance of dedicated ALOs in cancer care has been previously reported, with research finding that ALOs improve outcomes for Aboriginal cancer patients and improve the cultural knowledge of their non-Aboriginal colleagues [53]; when the ALO role was expanded to include community engagement activities, it helped promote collaborative links between services [32]. Stakeholders report that since the interviews, the ALO role at the AWCCC has experienced turnover, reflecting the challenges of recruiting to and retaining staff in a demanding role. Telehealth improved communication between services, as well as between services and patients, helping to build relationships between primary and tertiary services. Previous studies have found telehealth benefited rural health professionals through increased support from and access to specialists and via the experiential learning that occurs when the rural health professional is present for the specialist consultation, as well as improving networking and collaboration between rural clinicians and specialists [54,55]. However, although telehealth use has increased substantially in Australia and around the world as a direct result of the COVID-19 pandemic [56,57], many challenges to the successful use of telehealth in the NT remain [58]. A large study across remote communities in northern Australia found that while telehealth reduced patient travel and allowed PHC staff to participate in specialist appointments, it was resource intensive for already short-staffed clinics, requiring additional staff including interpreters to support telehealth appointments, and generating additional administrative work for staff [58]. Reliable and affordable internet connections are vital for effective telehealth in remote communities and we endorse the suggestion of training Aboriginal staff as digital navigators to ensure culturally safe telehealth consultations [58].

Although this study focused on improving communication between health services providing care for Aboriginal people with cancer, respectful, patient-centered care must be underpinned by cultural safety. While not a finding of this study, culturally safe communication between clinicians and patients is essential to improving health outcomes for Aboriginal peoples [59,60]. Evidence consistently shows that effective communication improves Indigenous engagement with health care services [61,62,63], with patients who received timely and clear communication about their cancer and treatment reporting more positive experiences of care [64,65].

### 4.1. Strengths and Limitations

This study provides insights into opportunities for improving the effectiveness of between-service communication to improve care in the NT for Aboriginal people with cancer. The number and remoteness of health professionals interviewed, including 14 Aboriginal staff among the 50 staff from a diverse range of locations and professions, is an exceptional contribution, providing unique insights. However, all PHC clinics in this study were operated by NT Health, as we were unsuccessful in our efforts to obtain perspectives of staff working at ACCHS. Further investigation into the perspectives of ACCHS staff on between-service communication in the NT is required, although it seems unlikely the situation with communication would be better. While caution should be exercised in generalising the study results to services for Indigenous patients with cancer in other states or countries, the study highlights the underlying importance of communication between the levels of service provision including the importance of integrating information systems.

The data for this study was collected between 2016 and 2019, however data analysis and write up were delayed for some time due to the outbreak of COVID-19, subsequent disruptions to staffing, communication and workload in the NT, and due to changes in the project team. We are aware of new systems, organisational restructures and roles that have been implemented since the interviews which affect how between-service communication is conducted in the NT. However, between May and June 2025 our findings were presented to a range of regional and remote stakeholders providing cancer care to Aboriginal people in the NT who validated the findings and confirmed that they are still relevant in the current environment.

Another limitation is that transcripts were not returned to participants for member checking. Incorporating this process may have strengthened the trustworthiness of the findings. Nonetheless, participants’ comments and experiences regarding between-service communication were repeated as analysis proceeded, indicating that saturation occurred and giving confidence to the conclusions reached.

### 4.2. Recommendations

There is much to be learned from health care providers perspectives on ways to improve between-service communication. We have developed a list of recommendations based on strategies suggested by interview participants (*n* = 50) and key stakeholders (*n* = 22) (Table 2). Although recommended strategies are listed under one primary finding, many strategies are relevant across several findings.

## 5. Conclusions

Good communication and strong relationships between PHC services, hospitals and specialists are critical to improving cancer care for Aboriginal people. Multiple strategies are outlined by which communication and the sharing of patient information between services can be enhanced. The enablers of communication need to be strengthened, including through designated staff in PHC clinics to support patients with cancer, dedicated Aboriginal cancer roles and additional resourcing to coordinate telehealth appointments. More timely communication between services, involving remote clinicians in cancer multidisciplinary teams (MDTs) and education for PHC staff on cancer requires barriers to communication be addressed. A renewed commitment by the NT government to improve the interoperability of health care IT systems in the NT and health policies that require timely information exchange and creating opportunities for strong partnerships between hospitals and PHC clinics can improve the communication and coordination of cancer care for Aboriginal people in the NT.

## Figures and Tables

**Figure 1 cancers-17-03155-f001:**
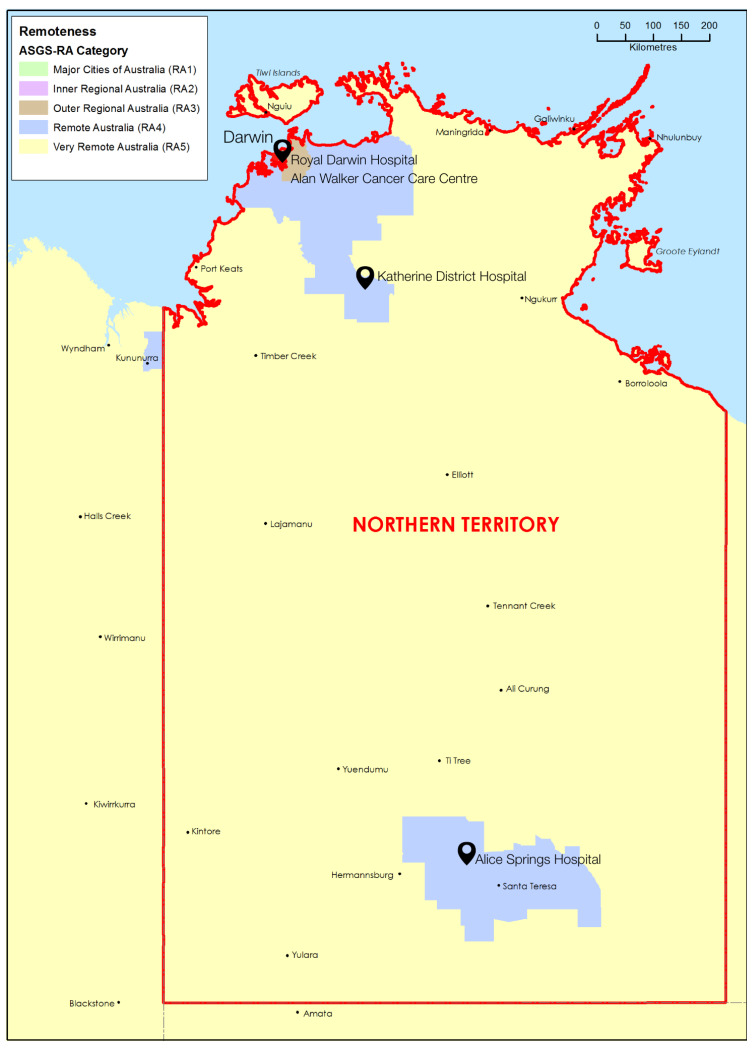
Location of public cancer services in the Northern Territory with Australian Statistical Geography Standard (ASGS) remoteness area classifications. Based on Commonwealth of Australia (Department of Health, Disability and Ageing) material [36].

**Figure 2 cancers-17-03155-f002:**
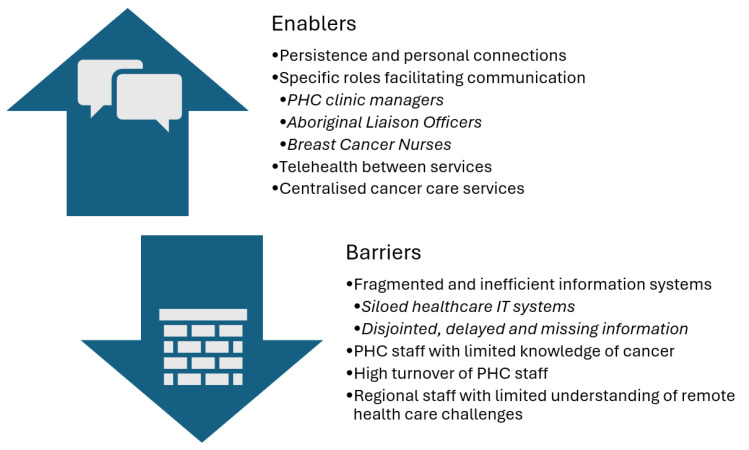
Barriers and enablers to communication between health services.

**Table 1 cancers-17-03155-t001:** Characteristics of health service staff.

Characteristics	Number of Staff (*n* = 50)	Proportion
**Remoteness**		
Outer regional	14	28%
Remote	13	26%
Very Remote	23	46%
**Gender**		
Female	38	76%
Male	12	24%
**Indigeneity**		
Aboriginal	14	28%
Non-Aboriginal	36	72%
**Employer**		
Aged care	1	2%
Cancer centre	6	12%
Cancer support service	1	2%
Hospital	7	14%
Palliative care	2	4%
Primary health care clinic	33	66%

**Table 2 cancers-17-03155-t002:** Participant and stakeholder recommended strategies to improve communication between health services.

Finding	Recommendation
Siloed Health Care IT Systems	Improved health care IT systems to readily facilitate sharing information between services. ^ Use and sharing of electronic patient records between hospital and PHC sites in real time. ^
Disjointed, delayed and missing information	More timely communication between services by telephone and e-mail. ^ Involve remote clinicians in cancer multidisciplinary teams (MDTs) and get their input for their patients. * Hospitals and specialists improve their timeliness of communication with PHC clinics about patient’s: ^ ○ Required investigations (blood tests and scans) ○ Diagnosis and probable care pathway ○ Upcoming appointments ○ Treatment while in hospital (if patient consents) ○ Requirements for care in their home/community (including medication and equipment) before discharge Contact details of hospital clinician on all discharge summaries; copy sent with patient and directly to clinic at time of discharge.
PHC staff with limited knowledge of cancer	Educate PHC staff on cancer, including on the specific pathway and who to contact in the hospital for more information for individual patients. Mentoring and training for remote Aboriginal staff about cancer. *
High turnover of PHC staff	More long-term staffing for PHC clinics, designating a staff member to support patients with cancer, promote cancer screening and provide cancer education. ^ Designated contact staff member at each site for health professionals to discuss patient care. *
Regional staff with limited understanding of remote health care challenges	Provide regional hospital staff opportunities to learn about health care at remote clinic sites (challenges and realities), including through cultural immersion experiences. *
Specific roles to enhance communication	Ensure access to dedicated Aboriginal cancer support roles in all regional treatment centres. ^ Increase the number of Aboriginal health practitioners to support cancer patients. *
Telehealth between services	Additional staff and resources to coordinate telehealth between services. * Case conferences between services using telehealth. *

^ Recommendations that were made both by the original participants and the key stakeholders. * Recommendations that were only provided by key stakeholders involved with cancer care in the NT.

## Data Availability

The datasets generated and/or analysed during the current study are not publicly available due to small participant numbers and protection of confidentiality. Aggregate data is available from the corresponding author on reasonable request.

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
