# Peer review of "“We Just Get Whispers Back”: Perspectives of Primary and Hospital Health Care Providers on Between-Service Communication for Aboriginal People with Cancer in the Northern Territory"

_cancers, 2025, doi:10.3390/cancers17193155_

Round 1
Reviewer 1 Report
Comments and Suggestions for Authors
Thanks to authors for a manuscript exploring the “perspectives of Northern Territory primary and hospital care providers” (lines 105-106, more on this below) on issues related to the challenges in communication during the provision of cancer care to Aboriginal patients living in remote areas of this Australian state using what seems to be a ‘qualitative study design’. The following points will improve this manuscript.
1. Lines 1-57: The title and the abstract should clearly indicate the study design used.
2. Lines 38-41: The methods section of the abstract should be clear about important dates and succinctly summarize all of the methods used in this study.
3. Lines 83-104: As authors indicate, there seem to be studies in the literature that are related to the challenges in communication and information flow in remote cancer care. Could authors discuss the gap they identified in these studies that prompted them to conduct this study in general? Also, please provide reasons what were the characteristics of this gap that made them choose what seems to be a ‘qualitative study design’ for their work.
4. Lines 105-108: How does the “perspectives of Northern Territory primary and hospital care providers” which seems to be the focus of this study help in “exploring strategies to improve communication…”? (More on this below.)
5. Line 110: Please provide a complete methods section so that the processes used in this study is explained precisely. It is important that the methods used in this study are totally transparent for the readership without asking them to go to another publication.
6. Line 110: Please also be transparent about the subject matter and relationship of the previous publication with this one.
7. Lines 110-182: Thanks to authors for providing some information (or all?) about methods used in this study. Given the reference to a previous publication, it is unclear if all of the processes used are completely and precisely laid out here. Therefore, this reviewer is not sure how to continue with their review of this section.
8. Lines 134-135: It is reported that the interviews were conducted between “September 2016 and April 2019” and now it is August 2025. Thanks to authors for indicating later that the “[d]ata analysis was delayed due to outbreak of COVID-19…” (lines 159-160) and that “[i]n 2025, the findings and recommendations were presented to 22 key stakeholders…” who were “commenting on certain aspects of cancer care delivery that had changed…” (lines 173-179), but, still, are authors content with their data that seems to be more than 6 years old by now and results that seem to have been ‘revised’ based on “22 stakeholders” input and, therefore, does no longer completely represent the “perspectives of Northern Territory primary and hospital care providers” (lines 105-106)?
9. Lines 157-160: If authors were unable to work on “data analysis” during “COVID-19” (which was declared a pandemic on “March 11, 2020” and, later, declared no longer a public health emergency on “May 5, 2023”, a la WHO and NIH websites), why the views of “22 key stakeholders” were sought in 2025? When in 2025?
10. Lines 173-182: Again, thanks for getting the views of “22 key stakeholders”. However, given that this manuscript reports on results from a ‘research study’, the main questions that arise over here are that, one, whose “perspectives” are being reported here and, two, what type of a research study design does this study correspond to?
Author Response
|
Overview comment:
Thanks to authors for a manuscript exploring the “perspectives of Northern Territory primary and hospital care providers” (lines 105-106, more on this below) on issues related to the challenges in communication during the provision of cancer care to Aboriginal patients living in remote areas of this Australian state using what seems to be a ‘qualitative study design’. The following points will improve this manuscript. |
We thank the reviewer for these comments.
|
|
Comment 1: 1. Lines 1-57: The title and the abstract should clearly indicate the study design used.
|
Response 1: We made the following change to the Simple Summary and the Abstract to clearly indicate that a qualitative approach was used:
Simple Summary: This qualitative study interviewed 50 staff from 15 health services across the NT and found that poor or delayed communication between health services negatively affected patient care and support.
Abstract: Methods: A qualitative study was undertaken in which semi-structured interviews were conducted with fifty staff from 15 health services (8 regional, remote, and very remote PHC clinics; 3 hospitals; one cancer centre and 3 cancer support services) between 2016 and 2019.
We feel that the title is appropriate and adequately represents the approach and content of the article. Many articles do not specifically indicate the study design in the title. |
|
Comment 2: 2. Lines 38-41: The methods section of the abstract should be clear about important dates and succinctly summarize all of the methods used in this study.
|
Response 2: We have added the dates to the Methods section of the abstract:
A qualitative study was undertaken in which semi-structured interviews were conducted with fifty staff from 15 health services (8 regional, remote, and very remote PHC clinics; 3 hospitals; one cancer centre and 3 cancer support services) between 2016 and 2019. |
|
Comment 3: 3. Lines 83-104: As authors indicate, there seem to be studies in the literature that are related to the challenges in communication and information flow in remote cancer care. Could authors discuss the gap they identified in these studies that prompted them to conduct this study in general? Also, please provide reasons what were the characteristics of this gap that made them choose what seems to be a ‘qualitative study design’ for their work.
|
Response 3: We have added the following to our Introduction:
Minimal research has explored the perspectives of both primary health care providers based in remote and very remote locations and hospital health care providers on between-service communication, particularly for Aboriginal cancer patients. Primary care and hospital-based health professionals, with their knowledge of the health system’s inner workings and complexities, are well placed to identify gaps in communication and cancer care coordination, as well as opportunities for improvement. In addition, health professionals working in remote areas have a unique understanding of the complexities of providing health care in a remote community, and the additional challenges of communicating with their hospital-based colleagues. To our knowledge, no such study has been conducted in the NT. This qualitative study was undertaken to investigate the perspectives of primary care providers based in regional and remote communities of the Northern Territory as well as the perspectives of hospital-based health care providers on communication issues between these services. It also reports recommended strategies to improve communication and enhance the delivery of cancer care for Aboriginal people. |
|
Comment 4: 4. Lines 105-108: How does the “perspectives of Northern Territory primary and hospital care providers” which seems to be the focus of this study help in “exploring strategies to improve communication…”? (More on this below.)
|
Response 4: We have added the following explanation of why we believe the perspectives of primary and hospital-based health professionals can help in exploring strategies to improve communication between their services.
Primary care and hospital-based health professionals, with their knowledge of the health system’s inner workings and complexities, are well placed to identify gaps in communication and cancer care coordination, as well as opportunities for improvement. In addition, health professionals working in remote areas have a unique understanding of the complexities of providing health care in a remote community, and the additional challenges of communicating with their hospital-based colleagues.
|
|
Comment 5: 5. Line 110: Please provide a complete methods section so that the processes used in this study is explained precisely. It is important that the methods used in this study are totally transparent for the readership without asking them to go to another publication.
|
Response 5: Additional details and project background is now provided in an expanded methods section to include the additional information requested.
We have added the following paragraph to section 2.1 Ethics approval to explain the project background:
This study was started as part of a national initiative aimed at identifying and describing cancer services providing treatment to Aboriginal cancer patients across Australia. It was a component of research conducted under the auspices of the Centre for Research Excellence (CRE), Discovering Indigenous Strategies to improve Cancer Outcomes Via Engagement, Research Translation and Training (DISCOVER-TT) funded by the National Health and Medical Research Council (NHMRC). The CRE was led by an Aboriginal researcher and brought together Aboriginal and non-Aboriginal researchers from around Australia, with the aim of improving outcomes and services for Aboriginal people with cancer. The study was carried out in accordance with the NHMRC Guidelines for Ethical Conduct in Aboriginal and Torres Strait Islander Health Research [34].
We also added the following paragraph to section 2.3 Participants and recruitment:
Potential staff participants were invited to take part in the study through personal approaches in clinical service areas and at staff meetings. Study information was provided either in-person or via email. Some participants forwarded the study information to colleagues or recommended additional participants to the research team. As data collection progressed and gaps in representation were identified, recruitment efforts shifted to focus on including participants whose perspectives were underrepresented. Participation was voluntary and all participants provided written or oral consent before being interviewed.
|
|
Comment 6: 6. Line 110: Please also be transparent about the subject matter and relationship of the previous publication with this one.
|
Response 6: We have provided the following paragraph at the start of our Methods section to clarify the subject matter and the relationship of the previous publication with this one. This study employed a qualitative design, drawing on semi-structured interviews with health professionals. Participants were recruited from PHC services in regional, remote and very remote geographical locations, as well as from hospital and cancer services in regional and remote locations. This qualitative methodology facilitated an in-depth exploration of health professionals’ experiences and perspectives on the provision of cancer care to Aboriginal patients. An article reporting on health care providers’ perspectives on provision of cancer screening to Aboriginal people in the NT has previously been published from this research study [33]. |
|
Comment 7: 7. Lines 110-182: Thanks to authors for providing some information (or all?) about methods used in this study. Given the reference to a previous publication, it is unclear if all of the processes used are completely and precisely laid out here. Therefore, this reviewer is not sure how to continue with their review of this section.
|
Response 7: We have now expanded the methods section to include some additional details and project background. |
|
8. Lines 134-135: It is reported that the interviews were conducted between “September 2016 and April 2019” and now it is August 2025. Thanks to authors for indicating later that the “[d]ata analysis was delayed due to outbreak of COVID-19…” (lines 159-160) and that “[i]n 2025, the findings and recommendations were presented to 22 key stakeholders…” who were “commenting on certain aspects of cancer care delivery that had changed…” (lines 173-179), but, still, are authors content with their data that seems to be more than 6 years old by now and results that seem to have been ‘revised’ based on “22 stakeholders” input and, therefore, does no longer completely represent the “perspectives of Northern Territory primary and hospital care providers” (lines 105-106)?
|
We acknowledge that the delay between data collection and publication in this study is not ideal. Reasons for delays in this study were multifactorial and included: · Time taken to get to know, work with and gain the trust of the health service personnel, particularly staff working in remote and very remote primary healthcare clinics. · Turnover of research team members · Covid-19, as staff were working in states affected by lockdown for all of 2020 and 2021. Attempts to re-engage NT staff were initially delayed by the Covid outbreak in the NT in late 2021 and early 2022. These reasons were mostly outside of our control and could not easily have been averted.
However, we wanted to honour the contributions made by the 50 health personnel who participated in this study so that their insights were not lost.
Due to the age of our data, once our analysis was complete, we presented our findings to 22 key stakeholders. They confirmed that the issues identified are still very relevant and the recommendations made in this article are important for policy understanding and refinement.
The results were not revised by the stakeholders and do not include the data from the stakeholders, although it affirmed the recommendations made. We did capture some additional contemporary recommendations; however, this is clearly indicated.
We have modified the Data Analysis section as follows:
Between May and June 2025, the findings and recommendations were presented to 22 key stakeholders to confirm their contemporary relevance. Regional, remote and very remote clinicians (n=11) involved with cancer care for Aboriginal people in the NT were approached and invited to provide feedback on our findings. In addition, our findings were presented to a team meeting of health service managers (n=11). After presenting our findings to the stakeholders their feedback and reactions to the findings were captured through extensive note taking. The stakeholders agreed with the identified barriers, enablers and recommendations and confirmed that the findings were still highly relevant. The views of the stakeholders are not reported in the Results section. In addition, the stakeholders were asked whether they had any additional recommendations to improve between-service communication, not already identified in our findings. The stakeholder recommendations were mapped to the existing recommendations, and new strategies were grouped according to which barrier or enabler they addressed. Recommendations based on strategies suggested by interview participants and key stakeholders are reported in the Discussion section 4.2 of this paper and any recommendations solely provided by the stakeholder group are identified.
|
|
9. Lines 157-160: If authors were unable to work on “data analysis” during “COVID-19” (which was declared a pandemic on “March 11, 2020” and, later, declared no longer a public health emergency on “May 5, 2023”, a la WHO and NIH websites), why the views of “22 key stakeholders” were sought in 2025? When in 2025?
|
Response 9: The views of the stakeholders were sought in 2025 because that is when the analysis was complete and the data was ready for presentation. We have clarified in the paper when in 2025 the presentations to the stakeholders took place and the rationale for including their feedback. |
|
10. Lines 173-182: Again, thanks for getting the views of “22 key stakeholders”. However, given that this manuscript reports on results from a ‘research study’, the main questions that arise over here are that, one, whose “perspectives” are being reported here and, two, what type of a research study design does this study correspond to?
|
Response 10: Our apologies that this was not clear in our original manuscript. We have clarified that the views of the 22 stakeholders were sought to confirm the contemporary relevance of our findings. The only section which included stakeholder perspectives was the Recommendations, which we have moved to the Discussion section 4.2 and we have added the following to the Data Analysis section:
The views of the stakeholders are not reported in the Results section. In addition, the stakeholders were asked whether they had any additional recommendations to improve between-service communication, not already identified in our findings. The stakeholder recommendations were mapped to the existing recommendations, and new strategies were grouped according to which barrier or enabler they addressed. Recommendations based on strategies suggested by interview participants and key stakeholders are reported in the Discussion section 4.2 of this paper and any recommendations solely provided by the stakeholder group are identified.
|
Reviewer 2 Report
Comments and Suggestions for Authors
Thank you for the opportunity to review this interesting and important paper, which explores inter-service communication in integrated care for Indigenous people with cancer in Australia's Northern Territory. This is a population which is triply-disadvantaged in being geographically underserved and having significant health inequalities through socioeconomic disadvantage and ethnic disparities, so effective care is vital for bridging these gaps; highlighting issues which impact care, as this paper does, is an important contribution. The research conducted is robust, methodology well explained, clear findings, a nice sense of participant voice, and clear recommendations.
There are only a few minor points of revision I would suggest:
- The authors use "Aboriginal" respectfully throughout for Aboriginal & Torres Strait Islanders; I would suggest that perhaps "Indigenous" as terminology might have better translatability internationally, as the issues discussed here likely have crossover in e.g. Canada, Scandinavia, USA etc with Indigenous populations there. If "Aboriginal" is preferred this should be better justified/explained.
- Figure 1 shows the remoteness well and contextualises the issues for the reader. Can it come earlier, before the methods?
- One thing I found lacking from the findings was any exploration of the Indigenous cultural elements, aside from one recommendation; while the barriers identified are very real and vividly explored, they could be describing any remote service. Was anything identified in the data which spoke specifically to the challenges of communicating to provide integrated care that is culturally-competent to the needs of Indigenous patients' cultural needs and values? Adding it in as a recommendation suggests there was something there, can this be reflected in the findings somehow also?
- The discussion could also reflect on how these findings might link with similar international settings where there are reported challenges with providing integrated care in remote regions with Indigenous populations, to increase the international utility (e.g. Pinero de Plaza et al have a strong systematic review of this https://ijic.org/articles/10.5334/ijic.7643)
- Check reference list for consistent linking of DOIs
Author Response
|
Overview comment: Thank you for the opportunity to review this interesting and important paper, which explores inter-service communication in integrated care for Indigenous people with cancer in Australia's Northern Territory. This is a population which is triply-disadvantaged in being geographically underserved and having significant health inequalities through socioeconomic disadvantage and ethnic disparities, so effective care is vital for bridging these gaps; highlighting issues which impact care, as this paper does, is an important contribution. The research conducted is robust, methodology well explained, clear findings, a nice sense of participant voice, and clear recommendations. There are only a few minor points of revision I would suggest:
|
We thank the reviewer for these comments.
|
|
Comment 1: 1. The authors use "Aboriginal" respectfully throughout for Aboriginal & Torres Strait Islanders; I would suggest that perhaps "Indigenous" as terminology might have better translatability internationally, as the issues discussed here likely have crossover in e.g. Canada, Scandinavia, USA etc with Indigenous populations there. If "Aboriginal" is preferred this should be better justified/explained.
|
Response 1: Terminology choice is a contested area and there is a diversity of opinions on preferred terminology. While we acknowledge that “Indigenous” is more recognised internationally, “Aboriginal” is the preferred term in the Northern Territory and the term “Indigenous” is often deemed inappropriate. We have added the following explanation to our Introduction:
Aboriginal and Torres Strait Islander peoples in Australia experience poorer cancer outcomes compared to other Australians. ‘Aboriginal’ is the term preferred in the Northern Territory and therefore is used in this paper to refer to the Aboriginal and Torres Strait Islander people who are traditional inhabitants and Indigenous people of Australia. 'Indigenous' has been used when we refer to the First Nations inhabitants of other countries. |
|
Comment 2: 2. Figure 1 shows the remoteness well and contextualises the issues for the reader. Can it come earlier, before the methods?
|
Response 2: We thank the reviewer for their comments on Figure 1. Upon reflection we have chosen to leave Figure 1 in the Methods section as we feel it flows well coming directly after the description of how we categorised the geographic remoteness of the NT health services. We are also concerned it might interrupt the flow if we included it in the introduction.
|
|
Comment 3: 3. One thing I found lacking from the findings was any exploration of the Indigenous cultural elements, aside from one recommendation; while the barriers identified are very real and vividly explored, they could be describing any remote service. Was anything identified in the data which spoke specifically to the challenges of communicating to provide integrated care that is culturally-competent to the needs of Indigenous patients' cultural needs and values? Adding it in as a recommendation suggests there was something there, can this be reflected in the findings somehow also?
|
Response 3: The reviewer makes an excellent point about the importance of cultural elements in the provision of care. While there was a lot of data in the interviews about providing culturally competent care that meets the needs of Aboriginal patients, this paper focusses on communication between services rather than between patients and health providers and cultural safety per se.
The data that was relevant to between-service communication was focused on the challenges of remote health care delivery, highlighting that when between-service communication didn’t occur (or didn’t occur well), Aboriginal patients are often more disadvantaged due to many factors which include their remoteness from the services.
To avoid confusion, we have renamed the finding in the Recommendations table to “Specific roles to enhance communication” to better align with the identified enabler in the Results section. While this enabler doesn’t have a strong cultural element, it does highlight the importance of Aboriginal Liaison Officers (ALOs) as a central, visible point of contact between the hospital and clinics. The recommendations that spoke to this finding suggested increasing the number of Aboriginal health professionals.
We have also added the following paragraph to the Discussion, acknowledging the importance of culturally safe communication between clinicians and patients:
Although this study focused on improving communication between health services providing care for Aboriginal people with cancer, respectful, patient-centered care must be underpinned by cultural safety. While not a finding of this study, culturally safe communication between clinicians and patients is essential to improving health outcomes for Aboriginal peoples [58,59]. Evidence consistently shows that effective communication improves Indigenous engagement with healthcare services [60-62], with patients who received timely and clear communication about their cancer and treatment reporting more positive experiences of care [63,64].
|
|
Comment 4: 4. The discussion could also reflect on how these findings might link with similar international settings where there are reported challenges with providing integrated care in remote regions with Indigenous populations, to increase the international utility (e.g. Pinero de Plaza et al have a strong systematic review of this https://ijic.org/articles/10.5334/ijic.7643)
|
Response 4: We have added multiple additional references to findings from international settings to our Discussion to highlight the potential for application of the findings to other remote regions or Indigenous contexts. This includes references to Pinero de Plaza et al. We give some examples below:
Staff gave numerous examples along the treatment pathway where communication breakdowns or delays led to patients falling through the gaps, being “lost” to follow up or disengaging from treatment, an issue that has long been reported both in Australia [40] and internationally [28,41].
An international narrative review found enablers for achieving continuity of care for First Nations people included information shared among providers and settings to ensure “collective memory”, links and referral strategies for care professionals and shared synchronized care records [38].
Around the world, the role of primary care in cancer diagnosis and management is growing [46,47]. Cancer is complex to manage, with numerous pathways depending on cancer type and stage [47] and these complexities are exacerbated for remote PHC staff. Criticisms that some PHC clinicians weren’t well informed of the appropriate cancer pathway or how to care for cancer patients adversely impacting care to their patients may be justified but reflects a failure of communication and issues related to staffing, including training, retention and support. This mirrors findings from an Ethiopian study on the coordination of cervical cancer care, which reported that lack of feedback on referred patients hampered learning for PHC providers and secondary hospitals, as they were unsure if their cancer screening referrals were appropriate [28].
|
|
Comment 5: 5. Check reference list for consistent linking of DOIs
|
Response 5: We have updated the reference list with consistent linking of DOIs. |
Reviewer 3 Report
Comments and Suggestions for Authors
The manuscript focuses on an interesting subject in the field of healthcare, i.e., the perspective of primary and hospital health care professionals on communication for Aboriginal people with cancer. The study brings important evidence on healthcare for individuals in remote areas, particularly vulnerable communities like Aboriginals, and is very quite nicely described, my recommendations concentrate on suggestions for improvement of the manuscript:
(1) Clarification of certain sentences in the study: there are some phrases that seem truncated or confusing (page 1, lines 25-29; page 2, lines 70-72);
(2) Use of parentheses in the text: the Introduction presents some explanations between parentheses that hinder readability of the text (page 2, lines 62-63 and line 74), maybe the explanations could be presented on footnotes or another sentence to improve the flow of ideas in the text;
(3) The transition between context and the objective of the study in the Introduction could be improved by emphasizing the gap in the literature and the contributions of the study in the field of knowledge;
(4) Furthermore, authors could take advantage of the importance of the study to extend the literature review to encompass evidence from minorities living in remote areas from other countries (e.g., Amazon region, African rural areas, small islands in the Caribbean and in the Eastern Asian regions, among others), highlighting in the Introduction and in the Discussion the potential for application of the findings on other challenging contexts;
(5) The initial sentence presenting the description of the study seem a little bit dismissive of readers ("Details of study methods can be found in a previous publication [20]", page 3, line 110), at least a paragraph describing the general context, study design and overall methods of the research could be presented before referencing previous work;
(6) The description of the data collection focuses on the interviews conducted between September 2016 and April 2019 (before the pandemics, pages 4-5), followed by presentation of further efforts to organize and incorporate additional contributions from stakeholders after the pandemics (page 6). Authors could clarify whether the "Fifty participants were interviewed from 15 health services" (page 6, line 184) include the "22 key stakeholders" interviewed in 2025 (page 6, line 173) or only the individuals interviewed between 2016 and 2019. In addition, it would be interesting to describe the criteria for selection of the health professionals interviewed in 2016-2019 and the stakeholders interviewed in 2025. Including a figure with the flowchart of the study phases and participants would also increase the interest of the paper;
(7) Figure 1 containing the map of the region included in the study is excellent, however, the caption of the map encompasses categories that are not included in the map (Major cities of Australia - RA1, and Inner Regional Australia - RA2);
(8) Table 1 should include the proportion of individuals in a third column (%).
Author Response
|
Overview comment: The manuscript focuses on an interesting subject in the field of healthcare, i.e., the perspective of primary and hospital health care professionals on communication for Aboriginal people with cancer. The study brings important evidence on healthcare for individuals in remote areas, particularly vulnerable communities like Aboriginals, and is very quite nicely described, my recommendations concentrate on suggestions for improvement of the manuscript:
|
We thank the reviewer for these comments.
|
|
Comment 1: (1) Clarification of certain sentences in the study: there are some phrases that seem truncated or confusing (page 1, lines 25-29; page 2, lines 70-72);
|
Response 1: We have attempted to clarify the identified sentences as follows.
Page 1: This research highlights the need for more timely communication and improved healthcare IT systems to enable better information sharing between services. It also identifies the need for designated clinic staff to support patients with cancer and dedicated Aboriginal cancer roles to improve the coordination and delivery of care for Aboriginal cancer patients.
Page 2: With 74% of Aboriginal people in the NT living in remote or very remote areas, the need to leave Country and travel long distances to access care in a health system which is not always seen as culturally secure presents well-recognised challenges [9,10].
|
|
Comment 2: (2) Use of parentheses in the text: the Introduction presents some explanations between parentheses that hinder readability of the text (page 2, lines 62-63 and line 74), maybe the explanations could be presented on footnotes or another sentence to improve the flow of ideas in the text; |
Response 2: We have removed the parentheses from the Introduction. In the first case the explanation has been presented in additional sentences. In general, health journals do not like the use of footnotes and so we have avoided these, and with the reworking we don’t feel any footnotes are necessary. Lines 62-63: Aboriginal and Torres Strait Islander peoples in Australia experience poorer cancer outcomes compared to other Australians; a disparity that has been observed in countries with Indigenous populations worldwide [1,2]. ‘Aboriginal’ is the term preferred in the Northern Territory and therefore is used in this paper to refer to the Aboriginal and Torres Strait Islander people who are traditional inhabitants and Indigenous people of Australia. 'Indigenous' has been used when we refer to the First Nations inhabitants of other countries.
Line 74: Delayed, absent or poor communication between healthcare professionals… |
|
Comment 3: (3) The transition between context and the objective of the study in the Introduction could be improved by emphasizing the gap in the literature and the contributions of the study in the field of knowledge;
|
Response 3: We have added the following to our Introduction:
Minimal research has explored the perspectives of both primary health care providers based in remote and very remote locations and hospital health care providers on between-service communication, particularly for Aboriginal cancer patients. Primary care and hospital-based health professionals, with their knowledge of the health system’s inner workings and complexities, are well placed to identify gaps in communication and cancer care coordination, as well as opportunities for improvement. In addition, health professionals working in remote areas have a unique understanding of the complexities of providing health care in a remote community, and the additional challenges of communicating with their hospital-based colleagues. To our knowledge, no such study has been conducted in the NT.
|
|
Comment 4: (4) Furthermore, authors could take advantage of the importance of the study to extend the literature review to encompass evidence from minorities living in remote areas from other countries (e.g., Amazon region, African rural areas, small islands in the Caribbean and in the Eastern Asian regions, among others), highlighting in the Introduction and in the Discussion the potential for application of the findings on other challenging contexts;
|
Response 4: We have added additional references from other countries to our Introduction. We have attempted to highlight in our Introduction that this is a global challenge. We give some examples below:
Cancer care is complex and often involves multiple different healthcare services, a complexity that increases with remoteness [11-13]. Delayed, absent or poor communication between healthcare professionals is a global challenge that may negatively impact continuity of care and patient outcomes [14,15]. Primary health care (PHC) clinics often act as health hubs in remote communities around the world and have an important role in cancer screening, diagnosis, and post-discharge cancer care, where patients are further from specialist cancer services, so accurate communication between hospitals, specialists and PHC clinics is vital [16-18]. Improving the coordination of cancer care is critical for addressing rural and remote patients’ needs and improving their cancer outcomes [19,20], and particularly salient for rural and remote Indigenous patients [21-23].
Challenges in coordinating care to meet Aboriginal cancer patients’ needs include lack of communication between services, delays in receiving timely hospital information, distance to treatment and provision of cultural support [10]. These challenges have also been reported by health service providers servicing rural communities in other parts of the world including in the United States of America (USA) [25-27], Ethiopia [28] and Scotland [29] and in rural and remote Indigenous communities in New Zealand [30], Canada [13] and the USA [31].
We have also added multiple additional references which internationalise our findings in our Discussion to highlight their potential application to other remote or Indigenous contexts. We give some examples below:
Staff gave numerous examples along the treatment pathway where communication breakdowns or delays led to patients falling through the gaps, being “lost” to follow up or disengaging from treatment, an issue that has long been reported both in Australia [40] and internationally [28,41].
An international narrative review found enablers for achieving continuity of care for First Nations people included information shared among providers and settings to ensure “collective memory”, links and referral strategies for care professionals and shared synchronized care records [38].
Around the world, the role of primary care in cancer diagnosis and management is growing [46,47]. Cancer is complex to manage, with numerous pathways depending on cancer type and stage [47] and these complexities are exacerbated for remote PHC staff. Criticisms that some PHC clinicians weren’t well informed of the appropriate cancer pathway or how to care for cancer patients, adversely impacting the care provided to patients may be justified but reflects a failure of communication and issues related to staffing, including training, retention and support. This mirrors findings from an Ethiopian study on the coordination of cervical cancer care, which reported that lack of feedback on referred patients hampered learning for PHC providers and secondary hospitals, as they were unsure if their cancer screening referrals were appropriate [28]. |
|
Comment 5: (5) The initial sentence presenting the description of the study seem a little bit dismissive of readers ("Details of study methods can be found in a previous publication [20]", page 3, line 110), at least a paragraph describing the general context, study design and overall methods of the research could be presented before referencing previous work;
|
Response 5: We have provided the following paragraph at the start of our Methods section to describe the general context, study and overall methods of the research before referencing previous work. This study employed a qualitative design, drawing on semi-structured interviews with health professionals. Participants were recruited from PHC services in regional, remote and very remote geographical locations, as well as from hospital and cancer services in regional and remote locations. This qualitative methodology facilitated an in-depth exploration of health professionals’ experiences and perspectives on the provision of cancer care to Aboriginal patients. An article reporting on health care providers’ perspectives on provision of cancer screening to Aboriginal people in the NT has previously been published from this research study [33].
We have also expanded the methods section to include some additional details and project background which was provided in the previous publication.
|
|
Comment 6: (6) The description of the data collection focuses on the interviews conducted between September 2016 and April 2019 (before the pandemics, pages 4-5), followed by presentation of further efforts to organize and incorporate additional contributions from stakeholders after the pandemics (page 6). Authors could clarify whether the "Fifty participants were interviewed from 15 health services" (page 6, line 184) include the "22 key stakeholders" interviewed in 2025 (page 6, line 173) or only the individuals interviewed between 2016 and 2019. In addition, it would be interesting to describe the criteria for selection of the health professionals interviewed in 2016-2019 and the stakeholders interviewed in 2025. Including a figure with the flowchart of the study phases and participants would also increase the interest of the paper;
|
Response 6: Our apologies that this was not clear in our original manuscript. We have clarified that the views of the 22 stakeholders were sought to confirm the contemporary relevance of our findings.
The results were not revised by the stakeholders and do not include data from the stakeholders. The only section which included stakeholder perspectives was the Recommendations, which we have now moved to the Discussion section 4.2.
Section 2.3 included the criteria for the selection of health professionals interviewed in 2016-2019, however in section 2.5 we have clarified the criteria for selecting the stakeholders.
We have modified the Data Analysis section as follows:
Between May and June 2025, the findings and recommendations were presented to 22 key stakeholders to confirm their contemporary relevance. Regional, remote and very remote clinicians (n=11) involved with cancer care for Aboriginal people in the NT were approached and invited to provide feedback on our findings. In addition, our findings were presented to a team meeting of health service managers (n=11). After presenting our findings to the stakeholders, their feedback and reactions to the findings were captured through extensive note taking. The stakeholders agreed with the identified barriers, enablers and recommendations and confirmed that the findings were still highly relevant. The views of the stakeholders are not reported in the Results section. In addition, the stakeholders were asked whether they had any additional recommendations to improve between-service communication, not already identified in our findings. The stakeholder recommendations were mapped to the existing recommendations, and new strategies were grouped according to which barrier or enabler they addressed. Recommendations based on strategies suggested by interview participants and key stakeholders are reported in the Discussion section 4.2 of this paper and any recommendations solely provided by the stakeholder group are identified.
|
|
Comment 7: (7) Figure 1 containing the map of the region included in the study is excellent, however, the caption of the map encompasses categories that are not included in the map (Major cities of Australia - RA1, and Inner Regional Australia - RA2);
|
Response 7: We thank the reviewer for their comments on Figure 1. Upon reflection we have left the additional categories “Major cities of Australia” and “Inner Regional Australia” on the map because we felt it was important to list the other categories that are used in Australia. This emphasises the remoteness of the NT, which does not have Major cities or areas classified as “inner regional”. The map is based with permission on Commonwealth of Australia (Department of Health, Disability and Ageing) material, and we have used the key that was provided with the map. |
|
Comment 8: (8) Table 1 should include the proportion of individuals in a third column (%).
|
Response 8: We have added a third column to Table 1 showing the proportion of individuals. |

Reviewer 4 Report
Comments and Suggestions for Authors
This is a well-written and useful article on how communications issues are present in cancer care for rural patients and it does not sufficiently describe or explain how data from a second cohort of interviewees were obtained and how their perspectives figured in the analysis. I make four broad suggestions about key sections of the paper. I think the authors can address these.
1) Please elaborate on the continuities and changes in the delivery system and the timeframes covered by descriptions of specific institutions. I think one the most important features of this paper is that you report on longitudinal data showing that despite ongoing efforts by the hospitals to improve communications, primary providers still see a strong need for improved communications. I would also consider searching for a few more references to communications related barriers to cancer care for rural patients from other countries as part of the introduction.
2) Please describe the a) acquisition and interviewing of the second recent sample of 22 people, b) what specific data was collected from them and how c) how this data was used in the the study, and d) add their characteristics to the table on study participants.
3) Please remove the cute graphics from the primary results figure 1. (And if it was intended to indirectly refer to cultural/racial-ethnic differences between hospital, district, and local staff and how they shape the communication, please say this more clearly.....or as a component of your findings on barriers to commuunication.)
4) Another strong aspect of the paper is the reporting of recommendations from (both the 2016-2019 cohort and the 2025 cohort. You need to add text to the methods section about how you solicted recommendations from each group and the approach you used in linking recommendations to specific concerns. The recommendations themselves could be described in a little more detail where possible. The table may need to be adjusted to clarify which recommendations came from which group or at both time frames and there should be a discussion of how intervening efforts might intersect with these.
Author Response
|
Overview comment:
This is a well-written and useful article on how communications issues are present in cancer care for rural patients and it does not sufficiently describe or explain how data from a second cohort of interviewees were obtained and how their perspectives figured in the analysis. I make four broad suggestions about key sections of the paper. I think the authors can address these.
|
We thank the reviewer for these comments.
|
|
Comment 1: 1 a) Please elaborate on the continuities and changes in the delivery system and the timeframes covered by descriptions of specific institutions. I think one the most important features of this paper is that you report on longitudinal data showing that despite ongoing efforts by the hospitals to improve communications, primary providers still see a strong need for improved communications.
1 b) I would also consider searching for a few more references to communications related barriers to cancer care for rural patients from other countries as part of the introduction.
|
Response 1 a: This was not a longitudinal study. We have modified the Methods section to ensure our approach is clear and explain the timeframes.
We acknowledge that the delay between data collection and publication in this study is not ideal. However, we wanted to honour the contributions made by the 50 health personnel who participated in this study so that their insights were not lost.
Due to the age of our data, once our analysis was complete, we presented our findings to 22 key stakeholders. They confirmed that the issues identified are still very relevant and the recommendations made in this article are important for policy understanding and refinement. The only section which included stakeholder perspectives was the Recommendations, which we have moved to the Discussion section 4.2.
Response 1b: We have added additional references from other countries to our Introduction. We have attempted to highlight in our Introduction that this is a global challenge. We give some examples below:
Cancer care is complex and often involves multiple different healthcare services, a complexity that increases with remoteness [11-13]. Delayed, absent or poor communication between healthcare professionals is a global challenge that may negatively impact continuity of care and patient outcomes [14,15]. Primary health care (PHC) clinics often act as health hubs in remote communities around the world and have an important role in cancer screening, diagnosis, and post-discharge cancer care, where patients are further from specialist cancer services, so accurate communication between hospitals, specialists and PHC clinics is vital [16-18]. Improving the coordination of cancer care is critical for addressing rural and remote patients’ needs and improving their cancer outcomes [19,20], and particularly salient for rural and remote Indigenous patients [21-23].
Challenges in coordinating care to meet Aboriginal cancer patients’ needs include lack of communication between services, delays in receiving timely hospital information, distance to treatment and provision of cultural support [10]. These challenges have also been reported by health service providers servicing rural communities in other parts of the world including in the United States of America (USA) [25-27], Ethiopia [28] and Scotland [29] and in rural and remote Indigenous communities in New Zealand [30], Canada [13] and the USA [31].
|
|
Comment 2: 2) Please describe the a) acquisition and interviewing of the second recent sample of 22 people, b) what specific data was collected from them and how c) how this data was used in the the study, and d) add their characteristics to the table on study participants.
|
Response 2: Our apologies if this was unclear in our original manuscript. We have clarified that the views of the 22 stakeholders were sought to confirm the contemporary relevance of our findings.
The results were not revised by the stakeholders and do not include data from the stakeholders. The only section which included stakeholder perspectives was in the Recommendations, which we have moved to Discussion section 4.2 and specifically confirms and augments the Recommendations. Therefore, we have not added their characteristics to the table on study participants.
In the Data Analysis section 2.5 we have clarified the criteria for selecting the stakeholders and how the data was collected and used as follows:
Between May and June 2025, the findings and recommendations were presented to 22 key stakeholders to confirm their contemporary relevance. Regional, remote and very remote clinicians (n=11) involved with cancer care for Aboriginal people in the NT were approached and invited to provide feedback on our findings. In addition, our findings were presented to a team meeting of health service managers (n=11). After presenting our findings to the stakeholders their feedback and reactions to the findings were captured through extensive note taking. The stakeholders agreed with the identified barriers, enablers and recommendations and confirmed that the findings were still highly relevant. The views of the stakeholders are not reported in the Results section. In addition, the stakeholders were asked whether they had any additional recommendations to improve between-service communication, not already identified in our findings. The stakeholder recommendations were mapped to the existing recommendations, and new strategies were grouped according to which barrier or enabler they addressed. Recommendations based on strategies suggested by interview participants and key stakeholders are reported in the Discussion section 4.2 of this paper and any recommendations solely provided by the stakeholder group are identified.
|
|
Comment 3: 3) Please remove the cute graphics from the primary results figure 1. (And if it was intended to indirectly refer to cultural/racial-ethnic differences between hospital, district, and local staff and how they shape the communication, please say this more clearly.....or as a component of your findings on barriers to communication.)
|
Response 3: We have removed the graphics from the Results Figure 2. |
|
Comment 4: 4) Another strong aspect of the paper is the reporting of recommendations from (both the 2016-2019 cohort and the 2025 cohort. You need to add text to the methods section about how you solicited recommendations from each group and the approach you used in linking recommendations to specific concerns. The recommendations themselves could be described in a little more detail where possible. The table may need to be adjusted to clarify which recommendations came from which group or at both time frames and there should be a discussion of how intervening efforts might intersect with these.
|
Response 4: To avoid confusion about who the study participants were and the involvement of the stakeholders, we have moved the recommendations table to the Discussion section 4.2. We have added additional information in the methods section about how we solicited recommendations from each group and how the recommendations were linked.
Recommended strategies were then grouped according to which barrier or enabler they primarily addressed, even though some strategies are relevant in addressing more than one barrier. Between May and June 2025, the findings and recommendations were presented to 22 key stakeholders to confirm their contemporary relevance. Regional, remote and very remote clinicians (n=11) involved with cancer care for Aboriginal people in the NT were approached and invited to provide feedback on our findings. In addition, our findings were presented to a team meeting of health service managers (n=11). After presenting our findings to the stakeholders their feedback and reactions to the findings were captured through extensive note taking. The stakeholders agreed with the identified barriers, enablers and recommendations and confirmed that the findings were still highly relevant. The views of the stakeholders are not reported in the Results section. In addition, the stakeholders were asked whether they had any additional recommendations to improve between-service communication, not already identified in our findings. The stakeholder recommendations were mapped to the existing recommendations, and new strategies were grouped according to which barrier or enabler they addressed. Recommendations based on strategies suggested by interview participants and key stakeholders are reported in the Discussion section 4.2 of this paper and any recommendations solely provided by the stakeholder group are identified.
In the Recommendations table, we have indicated which recommendations were made by both the original participants and the stakeholders, and which were just made by the stakeholders. |
Round 2
Reviewer 1 Report
Comments and Suggestions for Authors
Thanks to authors for revisions.
Points that need further attention are indicated below. Old numbering is used for convenience.
3. Old “Lines 83-104: As authors indicate, there seem to be studies in the literature that are related to the challenges in communication and information flow in remote cancer care. Could authors discuss the gap they identified in these studies that prompted them to conduct this study in general? Also, please provide reasons what were the characteristics of this gap that made them choose what seems to be a ‘qualitative study design’ for their work.”
Thanks for adding even more studies from the literature in this round. What exactly is the gap in these studies? Why did authors attempt to use a ‘qualitative study design’? (Other portions of the introduction that seem to represent new additions in this second round that are in red were mostly there in the first round of review.)
5. Old “Line 110: Please provide a complete methods section so that the processes used in this study is explained precisely. It is important that the methods used in this study are totally transparent for the readership without asking them to go to another publication.”
Thanks for revisions. Unfortunately, many of the methodological necessities of conducting a qualitative study [such as the guidelines used in reporting this study, a full description of the “semi-structured interview guide” development process, were there debriefings after interviews, were there iterative refinements of the guide accordingly, how the conduct of some interviews “at the participant’s workplace” and others “via telephone” (now lines 178-180) affected their responses, what was done to ensure data saturation (new lines 162-174), what are the initials of the “highly experienced remote area nurse” (new lines 177-178), what was the exact process that was used in the transcription phase by those “two team members” and which team members are those (new lines 184-185), what about discrepancies in the transcription phase, what was the exact process in data coding and in which exact manner did those “two researchers” (new lines 211-212) “re-examined” data (if they were focused on different subject matters as discussed, unfortunately, that defeats the purpose of having two coders in the first place), why were there further need for two other researchers to “further review” codes (new lines 215-217), still it is not clear the exact manner with which these “further reviews” were conducted, how many codes were developed, was there a participant check…] are missing.
7. Old “Lines 110-182: Thanks to authors for providing some information (or all?) about methods used in this study. Given the reference to a previous publication, it is unclear if all of the processes used are completely and precisely laid out here. Therefore, this reviewer is not sure how to continue with their review of this section.”
Please see #5 above.
8. Old “Lines 134-135: It is reported that the interviews were conducted between “September 2016 and April 2019” and now it is August 2025. Thanks to authors for indicating later that the “[d]ata analysis was delayed due to outbreak of COVID-19…” (lines 159-160) and that “[i]n 2025, the findings and recommendations were presented to 22 key stakeholders…” who were “commenting on certain aspects of cancer care delivery that had changed…” (lines 173-179), but, still, are authors content with their data that seems to be more than 6 years old by now and results that seem to have been ‘revised’ based on “22 stakeholders” input and, therefore, does no longer completely represent the “perspectives of Northern Territory primary and hospital care providers” (lines 105-106)?”
Thanks for revisions and indicating that “[t]he views of the stakeholders are not reported in the Results section” (new lines 230). This now confirms that the data used in this study is very (6 years) old.
9. Old “Lines 157-160: If authors were unable to work on “data analysis” during “COVID-19” (which was declared a pandemic on “March 11, 2020” and, later, declared no longer a public health emergency on “May 5, 2023”, a la WHO and NIH websites), why the views of “22 key stakeholders” were sought in 2025? When in 2025?”
It is still unclear why there has been this wait especially given that we do not live in a static world and that research projects with data as old as 6 years cannot maintain their credibility.
10. Old “Lines 173-182: Again, thanks for getting the views of “22 key stakeholders”. However, given that this manuscript reports on results from a ‘research study’, the main questions that arise over here are that, one, whose “perspectives” are being reported here and, two, what type of a research study design does this study correspond to?”
Thanks for indicating that results were presented to the “stakeholders to confirm their contemporary relevance” (new lines 222-223). If the stakeholders already had information about these topics, why all of this time and energy were spent to conduct this study?
Author Response
|
Thanks to authors for revisions. Points that need further attention are indicated below. Old numbering is used for convenience.
|
|
|
3. Original comment: “Lines 83-104: As authors indicate, there seem to be studies in the literature that are related to the challenges in communication and information flow in remote cancer care. Could authors discuss the gap they identified in these studies that prompted them to conduct this study in general? Also, please provide reasons what were the characteristics of this gap that made them choose what seems to be a ‘qualitative study design’ for their work.”
New comment: Thanks for adding even more studies from the literature in this round. What exactly is the gap in these studies? Why did authors attempt to use a ‘qualitative study design’? (Other portions of the introduction that seem to represent new additions in this second round that are in red were mostly there in the first round of review.) |
Response: The gap is the dearth of research that asks rural and remote primary health care professionals their opinions about between-service communication. There is also minimal research on how poor between-service communication affects Aboriginal patients. No research has been published on this topic in the Northern Territory. We believe the following paragraph (previously added) to the Introduction describes the gap in these studies:
Minimal research has explored the perspectives of both primary health care providers based in remote and very remote locations and hospital health care providers on between-service communication, particularly for Aboriginal cancer patients. Primary care and hospital-based health professionals, with their knowledge of the health system’s inner workings and complexities, are well placed to identify gaps in communication and cancer care coordination, as well as opportunities for improvement. In addition, health professionals working in remote areas have a unique understanding of the complexities of providing health care in a remote community, and the additional challenges of communicating with their hospital-based colleagues. To our knowledge, no such study has been conducted in the NT.
We used a qualitative study design to gain an in-depth understanding of a complex situation – to explore the perspectives of the health care providers caring for Aboriginal cancer patients. These perspectives and the rich data collected through interviews could not have been obtained through a quantitative study. The following sentence (previously added) to the Introduction explains why we used a qualitative study design.
This qualitative study was undertaken to investigate the perspectives primary care providers based in regional and remote communities of the Northern Territory as well as the perspectives of hospital-based health care providers on communication issues between these services.
|
|
5. Original comment: “Line 110: Please provide a complete methods section so that the processes used in this study is explained precisely. It is important that the methods used in this study are totally transparent for the readership without asking them to go to another publication.”
New comment: Thanks for revisions. Unfortunately, many of the methodological necessities of conducting a qualitative study are missing such as: a) the guidelines used in reporting this study b) a full description of the “semi-structured interview guide” development process, were there debriefings after interviews, were there iterative refinements of the guide accordingly c) how the conduct of some interviews “at the participant’s workplace” and others “via telephone” (now lines 178-180) affected their responses d) what was done to ensure data saturation (new lines 162-174) e) what are the initials of the “highly experienced remote area nurse” (new lines 177-178) f) what was the exact process that was used in the transcription phase by those “two team members” and which team members are those (new lines 184-185), what about discrepancies in the transcription phase g) what was the exact process in data coding and in which exact manner did those “two researchers” (new lines 211-212) “re-examined” data (if they were focused on different subject matters as discussed, unfortunately, that defeats the purpose of having two coders in the first place) h) why were there further need for two other researchers to “further review” codes (new lines 215-217), still it is not clear the exact manner with which these “further reviews” were conducted i) how many codes were developed j) was there a participant check
|
a) We have clarified this by adding the following to the Methods (Data Analysis) section: Reporting of the findings was guided by the consolidated criteria for reporting qualitative research (COREQ) checklist [38]; checklist available on request. b) We have clarified the interview development process by adding the following to the Data Collection section: A semi-structured interview guide was developed by the investigators, reviewed by the Aboriginal Advisory Group and refined. c) The majority of interviews were conducted at the participants workplace. A small number were conducted via telephone due to issues of staff availability and geographic distance. The small number of telephone interviews makes it hard to compare, but there were no observed differences between the responses based on the interview method. d) We have added the following to the Limitations section regarding data saturation: Another limitation is that transcripts were not returned to participants for member checking. Incorporating this process may have strengthened the trustworthiness of the findings. Nonetheless, participants’ comments and experiences regarding between-service communication were repeated as analysis proceeded, indicating that saturation occurred and giving confidence to the conclusions reached. e) The remote area nurse who conducted the interviews has since left the project and is not a co-author on the paper as despite considerable effort we were not able to contact her. Therefore, it is not appropriate to provide her initials. f) The majority of the transcription was performed manually by the interviewer, who has since left the team. EVT transcribed the remaining audio files and checked the existing transcriptions against the audio files to ensure accuracy. g) Initial coding was undertaken by EVT, as described in the paper. Codes were identified on a number of topics including many codes related to communication. During this analysis and discussion between the team, data related to between-service communication was identified as an area of interest and very relevant in the current context. EVT and AE did not re-examine the data with a different focus, instead they re-examined the existing codes with the purpose of focusing on the specific area of between-service communication; to develop and refine existing codes and make sure all data related to this topic was coded. We have clarified this point in the Data Analysis section as follows: Two researchers (EVT and AE) re-examined the existing themes to identify and refine themes specifically focused on between-service communication. h) The two additional reviewers reviewed a sample of the coded transcripts and the emerging themes related to between-service communication to ensure that themes were consistently applied and meaningful in the current context. The two additional reviewers contributed to refining the themes. We have clarified this point in the Data Analysis section as follows: BA and PM then reviewed a sample of the coded transcripts and the emerging themes related to between-service communication to ensure that themes were consistently applied and meaningful in the current context. i) Using Nvivo EVT initially generated 60 parent codes from the data. There were 12 initial codes specific to between-service communication. j) No there was not a participant check. We have added a comment in the Limitations section, where we state: Another limitation is that transcripts were not returned to participants for member checking. Incorporating this process may have strengthened the trustworthiness of the findings. Nonetheless, participants’ comments and experiences regarding between-service communication were repeated as analysis proceeded, indicating that saturation occurred and giving confidence to the conclusions reached. |
|
7. Please see #5 above.
|
See above. |
|
8. Thanks for revisions and indicating that “[t]he views of the stakeholders are not reported in the Results section” (new lines 230). This now confirms that the data used in this study is very (6 years) old.
|
Response: We have acknowledged that the delay between data collection and publication in this study is not ideal. However, due to the age of our data, once our analysis was complete, we presented our findings to 22 key stakeholders. They confirmed that the issues identified are still very relevant and the recommendations made in this article are important for policy understanding and refinement. Information such as this can be used to highlight issues and advocate for changes in resourcing and focus.
|
|
9. Original comment: “Lines 157-160: If authors were unable to work on “data analysis” during “COVID-19” (which was declared a pandemic on “March 11, 2020” and, later, declared no longer a public health emergency on “May 5, 2023”, a la WHO and NIH websites), why the views of “22 key stakeholders” were sought in 2025? When in 2025?”
New comment: It is still unclear why there has been this wait especially given that we do not live in a static world and that research projects with data as old as 6 years cannot maintain their credibility.
|
We acknowledged that the delay between data collection and publication in this study is not ideal. However, we wanted to honour the contributions made by the 50 health personnel who participated in this study so that their insights were not lost.
This study was originally being run by a PhD student. She conducted the interviews and started the transcription process. Unfortunately, she experienced personal and family problems which delayed the transcription process as she moved interstate. Then around the time of Covid-19 emerging she disengaged from her PhD and the project. Unfortunately, the project then did not progress for a long period while the university attempted to re-engage her. All of this took some time, especially as the COVID-19 outbreak occurred in the NT later than other parts of the world due to its remoteness. The NT had its peak outbreak in late 2021 and early 2022.
Eventually the project was handed to another researcher (EVT the lead author). However, by that stage funding had ceased, and that researcher was only working two days a week (and had other concurrent work commitments). The new researcher had to familiarize herself with the 50 interviews, complete transcription, analyse the data to understand what had been gathered, and then create a new team of researchers based in the NT to work with. Only the PhD supervisor (SCT) remained with the project from its outset. The new team members based in the NT are all busy clinicians who have met regularly, donating their time to help report on issues that they believed to be important.
Due to the delay in reporting the data, we had to be selective about which findings from the data we wrote up. The NT-based members of the research team provided guidance on which areas were still relevant and which we should focus our attention on. Initially, the decision was made to focus on cancer screening as this is still an issue in the NT. An article was written about the screening findings.
Then in 2024 the researchers turned their attention to between-service communication as there was rich data on this and it was identified by the NT co-authors to still be an issue in the NT. This is still a very real issue in the NT (and many other parts of Australia and the world).
We believe that the credibility of the data is maintained through presenting our findings to 22 key stakeholders. They confirmed that the issues identified are still very relevant and the recommendations made in this article are important for policy understanding and refinement. Information such as this can be used to highlight issues and advocate for changes in resourcing and focus.
|
|
10. Original comment: “Lines 173-182: Again, thanks for getting the views of “22 key stakeholders”. However, given that this manuscript reports on results from a ‘research study’, the main questions that arise over here are that, one, whose “perspectives” are being reported here and, two, what type of a research study design does this study correspond to?”
New comment: Thanks for indicating that results were presented to the “stakeholders to confirm their contemporary relevance” (new lines 222-223). If the stakeholders already had information about these topics, why all of this time and energy were spent to conduct this study?
|
This is not an issue that has been previously studied or reported in the NT. Therefore, since the stakeholders (and the teams they managed) had experienced identical issues to those identified in our analysis, they agreed that it was important that we publish the findings to help raise awareness of this issue.
It is important to note that there are sensitivities in Aboriginal health research and so it was particularly important to ensure that relevant people were involved in the consultations and feedback about the research and gave approval for the publication to be submitted. |
Reviewer 4 Report
Comments and Suggestions for Authors
I have one minor edit: lines 97-100 ..."united states of america" and "usa" reference the same place..
Overall, I think the paper is much stronger....you: 1) added more current and international references that address comparable issues; 2) clarified the use of the second group of 22 professionals and 3) treated the recommendations as discussion materials.
I think the paper is ready to go except for the item noted above and you have responded to reviewers' concerns. I think the paper underscores how much inadequate systems for clinical communications between facilities and levels of care can be major barriers to effective post-hospital care for complex cancer cases.
Author Response
|
Comment 1: I have one minor edit: lines 97-100 ..."united states of america" and "usa" reference the same place.. |
Thank you we have removed the second reference to the USA. The sentence now reads: These challenges have also been reported by health service providers servicing rural communities in other parts of the world including in the United States of America (USA) [25-28], Ethiopia [29] and Scotland [30] and in rural and remote Indigenous communities in New Zealand [31] and Canada [13]. |
|
Comment 2: Overall, I think the paper is much stronger....you: 1) added more current and international references that address comparable issues; 2) clarified the use of the second group of 22 professionals and 3) treated the recommendations as discussion materials. I think the paper is ready to go except for the item noted above and you have responded to reviewers' concerns. I think the paper underscores how much inadequate systems for clinical communications between facilities and levels of care can be major barriers to effective post-hospital care for complex cancer cases.
|
We thank the reviewer for these comments.
|